# Fusarium sacchari hypovirus 1, a Member of *Hypoviridae* with Virulence Attenuation Capacity in Phytopathogenic *Fusarium* Species

**DOI:** 10.3390/v16040608

**Published:** 2024-04-15

**Authors:** Qiujuan Zhou, Ziting Yao, Xueying Cao, Yuejia Chen, Chengwu Zou, Baoshan Chen

**Affiliations:** 1State Key Laboratory for Conservation and Utilization of Subtropical Agro-Bioresources and Ministry and Province Co-Sponsored Center of Collaborative Innovation for Sugarcane Industry, College of Life Science and Technology, Guangxi University, Nanning 530004, China; 2Plant Protection Research Institute, Guangxi Academy of Agriculture Science, Nanning 530007, China; 3Guangxi Key Laboratory of Sugarcane Biology, College of Agriculture, Guangxi University, Nanning 530004, China

**Keywords:** Pokkah boeng, *Fusarium* species, Fusarium sacchari hypovirus 1 (FsHV1), virulence

## Abstract

In a survey of mycoviruses in *Fusarium* species that cause sugarcane Pokkah boeng disease, twelve *Fusarium* strains from three *Fusarium* species (*F. sacchari*, *F. andiyazi*, and *F. solani*) were found to contain Fusarium sacchari hypovirus 1 (FsHV1), which we reported previously. The genomes of these variants range from 13,966 to 13,983 nucleotides, with 98.6% to 99.9% nucleotide sequence identity and 98.70% to 99.9% protein sequence similarity. Phylogenetic analysis placed these FsHV1 variants within the *Alphahypovirus* cluster of *Hypoviridae*. Intriguingly, no clear correlation was found between the geographic origin and host specificity of these viral variants. Additionally, six out of the twelve variants displayed segmental deletions of 1.5 to 1.8 kilobases, suggesting the existence of defective viral dsRNA. The presence of defective viral dsRNA led to a two-thirds reduction in the dsRNA of the wild-type viral genome, yet a tenfold increase in the total viral dsRNA content. To standardize virulence across natural strains, all FsHV1 strains were transferred into a single, virus-free *Fusarium* recipient strain, FZ06-VF, via mycelial fusion. Strains of *Fusarium* carrying FsHV1 exhibited suppressed pigment synthesis, diminished microspore production, and a marked decrease in virulence. Inoculation tests revealed varying capacities among different FsHV1 variants to modulate fungal virulence, with the strain harboring the FsHV1-FSA1 showing the lowest virulence, with a disease severity index (DSI) of 3.33, and the FsHV1-FS1 the highest (DSI = 17.66). The identification of highly virulent FsHV1 variants holds promise for the development of biocontrol agents for Pokkah boeng management.

## 1. Introduction

Pokkah boeng disease (PBD) is one of the most significant fungal diseases of sugarcane worldwide, seriously impacting sugarcane production [1]. Initially, the new leaves dechlorinate and turn yellow, and then the leaves gradually shrink and twist, harming the growth of the sugarcane. PBD is caused by the *Fusarium* species complex that includes *F. sacchari* [2,3], *F. andiyazi* [4], *F. oxysporum* [5], *F. verticillioides*, and *F. proliferatum* [6]. *Fusarium* is a genus of filamentous fungi of the phylum Ascomycota. These fungi are known to cause various symptoms such as blights, rots, cankers, and wilts in different parts of plants, affecting a wide range of crops [7,8]. While antifungal agents have shown some level of efficacy in managing PBD, *Fusarium* species quickly build up resistance to these treatments [9]. Mycoviruses, as a biocontrol agent, are a potential way to reduce the economic impact on sugarcane [10].

Mycoviruses are frequently encountered in plant pathogenic fungi, with double-stranded RNA (dsRNA) and positive-sense single-stranded RNA ((+) ssRNA) being the predominant types of genetic material [11]. Subsequently, single-stranded DNA (ssDNA) viruses have also been identified [12]. The majority of these mycoviruses are benign to their fungal hosts, meaning that they typically do not induce any phenotypic changes [13,14,15]. Nevertheless, an increasing body of evidence suggests that a growing number of mycoviruses can exert significant effects on their host fungi. Cryphonectria hypovirus 1 (CHV1), which infects *Cryphonectria parasitica* and attenuates fungal virulence, suppresses sporulation and pigmentation, and alters host gene expression patterns, is the best studied mycovirus [16]. To date, only a few mycoviruses have been reported as being capable of attenuating the virulence of their host fungi. These include Rosellinia necatrix megabirnavirus 1 (RnMBV 1) from the white root rot fungus *Rosellinia necatrix* [17], Sclerotinia sclerotiorum hypovirulence-associated DNA virus 1 (SsHADV-1) and Sclerotinia sclerotiorum hypovirus 2 (SsHV2/5472)from *Sclerotinia sclerotiorum*, a worldwide destructive fungal pathogen responsible for many economically important crops [18,19], and Hymenoscyphus fraxineus mitovirus 2 (HfMV2) from *Hymenoscyphus fraxineus*, a pathogen for ash dieback [20].

The *Fusarium* genus contains many phytopathogenic species and is responsible for several devastating diseases such as wheat white dead blight, maize fusarium wilt, and sugarcane Pokkah boeng in various important crops. To date, mycoviruses have been reported in 19 *Fusarium* species. *Mitoviridae* has the largest number of species (22 species), followed by *Partitiviridae* (12 species) and *Botourmiaviridae* (9 species) (Appendix A). Although diversified viruses are found in *Fusarium*, only three viruses have so far been reported to have a profound impact on their hosts, particularly the virulence and virulence-associated traits. Infection by Fusarium graminearum mycotymovirus 1 (FgMTV1/SX64) resulted in a reduction in growth rate and deoxynivalenol (DON) yield [21] and infection by Fusarium graminearum virus 1 (FgV1) suppressed sporulation and attenuated virulence in *F. graminearum* [22]. Infection by Fusarium oxysporum f. sp. dianthi virus 1 (FodV1) could cause a significant reduction in the vegetative growth and virulence of its fungal host *Fusarium oxysporum* f. sp. dianthi [23]. However, there is no report of mycoviruses capable of attenuating host virulence in the fungal pathogen *F. sacchari*.

*Hypoviridae* is a naked virus family with a positive single-stranded RNA genome ranging between 7.3 and 18.3 kb, typically containing one or two large open reading frames (ORFs). This family is currently composed of 8 genera, 39 accepted species [24], and 16 tentative species with their sequences deposited in the GenBank (Appendix A). These viruses are reported to be associated with 30 pathogenic fungi of economically significant plants, such as sugarcane, wheat, and rice. Although more members of the *Hypoviridae* family have been discovered, the biology of many of these viruses has not been adequately studied in general.

We previously identified the new mycovirus Fusarium sacchari hypovirus 1 (FsHV1) from *F. sacchari*, which belongs to the genus *Alphahypovirus* in the family *Hypoviridae*. It contains a single ORF (12,774 nt) encoding a polyprotein of 4258 amino acids and has 54% sequence homology with Wuhan insect virus 14 (WIV14) [25]. In this study, we assessed the genetic diversity and virulence regulation of 12 FsHV1 variants isolated from three *Fusarium* species, aiming to identify viral strains with biocontrol potential for Pokkah boeng disease.

## 2. Materials and Methods

### 2.1. Fungal Strains and Sugarcane Plants

*Fusarium* strains harboring only FsHV1 were used in this study (Appendix A). Virus-free strain FZ06-VF was obtained from the FZ06 strain by means of protoplast preparation and ultra-low detoxification (Appendix A).

The sugarcane variety Zhongzhe 9 (ZZ9), susceptible to sugarcane Pokkah boeng disease, was grown in a glasshouse and used for the fungal virulence assay.

### 2.2. Isolation and Identification of Fusarium Species

To isolate the *Fusarium* species, chlorotic and curled leaf tissues as well as sugarcanes showing Pokkah boeng symptoms were collected. The isolation of *Fusarium* from the leaves was performed as described by Bao et al. [2]. Soil samples were collected from the roots of sugarcane at a depth of about 10 cm and kept in sterile polythene bags. The isolation of the fungal strains was conducted within 12 h after sample collection. The dilution plate method for isolating *Fusarium* from rhizosphere soil was performed as described by Liu et al. [26], with modifications. Each dilution was spread onto PDA media supplemented with 100 mg/L chloramphenicol to inhibit Gram-positive bacteria and 100 mg/L streptomycin to inhibit Gram-negative bacteria. The plates were incubated in the dark at 28 °C for 2 to 3 days. Once the *Fusarium* species grew, the fungal mycelium was picked and transferred to fresh PDA for single-spore isolation and purification. Microconidia were observed on PDA, while macroconidia and sporulation patterns were observed on carnation leaf-piece agar (CLA).

The molecular identification of *Fusarium* species was performed following the method described previously [4]. PCR amplification was carried out using primers specific for three genes: the translation elongation factor 1-alpha (*TEF-1α*, EF1/EF2, 650 bp) gene [27], DNA-directed RNA polymerase II largest subunit (*RPB1*, Fa/G2R, 1.6 kb) gene [28], and the second-largest subunit of RNA polymerase II, *RPB2* (using two primer pairs: 5f2/7cr and 7cf/11ar, 1.8 kb) [29].

### 2.3. Extraction of DNA and RNA

The *Fusarium* strains were grown in 100 mL of potato dextrose water medium for 5 days and the mycelium was collected by means of filtration using Miracloth. Total nucleic acids were prepared using the method of Suzuki et al. [30] with modifications. The powder was transferred to a 2.0 mL centrifuge tube with 1000 μL of pre-filled extraction buffer (10 mL of 1 M Tris-HCl, 4 mL of 5 M NaCl, 5 mL of 80 mM EDTA pH 8.0, 20 mL of 10% SDS, and 61 mL of sterile water) and mixed with 1000 μL of RNA extraction phenol reagent (Solarbio, Beijing, China). The mixture was vortexed and centrifuged at 12,000 rpm for 10 min at room temperature. An aliquot of 800 μL of the supernatant was transferred to a new tube containing an equal volume of phenol–chloroform (1:1) mixture, vortexed, and centrifuged at 12,000 rpm for 10 min at 4 °C. This step was repeated twice more. Finally, 500 μL of the supernatant was mixed with 1500 μL of chilled absolute ethanol and stored at −80 °C for 1 h. The precipitate was washed three times with 200 μL of 75% ethanol and dissolved in RNase-free water. The nucleic acids were electrophoresed in a 1% agarose gel and observed after staining with ethidium bromide.

### 2.4. Purification of dsRNA, cDNA Synthesis, and RT-PCR

After the extraction of total RNA, the residual DNA was digested using RNase-free DNase Ι (TaKaRa, Dalian, China) at 37 °C for 30 min. The nucleic acids were then precipitated and treated with S1 Nuclease (TaKaRa, Dalian, China) at 37 °C for 30 min to remove single-stranded RNA and purify the dsRNA. The dsRNA was electrophoresed using low-melting agarose (SeaPlaque GTG, Rockland, ME, USA) and recovered using the FastPure Gel DNA Extraction Mini Kit (Vazyme, Nanjing, China). The dsRNAs were then analyzed using 1.2% agarose gel electrophoresis.

Reverse transcription was performed using TransScript One-Step gDNA Removal and cDNA Synthesis SuperMix (Transgen, Beijing, China). The total RNA (1.0 µg) was mixed with a random primer according to the manufacturer’s instructions for cDNA synthesis.

The full-length sequence of each FsHV1 variant was obtained by using RT-PCR and RACE with the SMARTer RACE 5′/3′ kit (Clonetech Laboratories, Mountain View, CA, USA) according to the manufacturer’s instructions. Primers 5′-GSP1/5′-GSP2 were used for 5′ RACE and primers 3′-GSP1/3′-GSP2 for 3′ RACE. The PCR amplicons were sequenced by Beijing Aoke Biotech Co. Ltd. (Beijing, China).

To quantify RdRp and defective fragments of the virus, quantitative PCR (qPCR) was performed using the cDNA of the *Fusarium* total RNA as a template with TransScript^®^ Green One-Step qRT-PCR SuperMix (TransGen Biotec Co., Ltd., Beijing, China) and LightCycler 480II (Roche, Basel, Switzerland). Primer pairs FsHV1-RdRpF1/FsHV1-RdRpR1 were used for the RdRp and FsHV1-DefectF/FsHV1-DefectR for the defective viral dsRNA. Primers used in this study are listed in Appendix A.

### 2.5. Sequence Assembly, Alignment, and Phylogenetic Analysis

The viral sequence assembly was performed using Vector NTI Advance 11.5.2. The gene sequences of *TEF-1α*, *RPB1*, and *RPB2* from species of *Fusarium* and the genome sequences of hypoviruses were obtained from the GenBank database (https://www.ncbi.nlm.nih.gov/genbank/, accessed on 1 April 2024). The phylogenetic tree of *Fusarium* species was constructed using *TEF-1α*, *RPB1*, and *RPB2* genes and the phylogenetic tree of Hypoviruses was constructed using the whole virus genomes. Multiple alignments were conducted using Clustal W in MEGA 7.0 (Molecular Evolutionary Genetic Analysis). The phylogenetic tree was constructed using the neighbor-joining (NJ) method with bootstrap resampling of 1000 replications to assess the reliability of the nodes in the phylogenetic tree.

### 2.6. Horizontal Transmission of the Viruses

A horizontal transmission assay was performed via mycelial fusion using the virus strains FS1, FS3, FS18, FS9, FS65, FS66, FSA1, LZ1, LZ8, LZ12, LZ14, and FZ06 as the donor and the virus-free *F. sacchari* strain FZ06-VF as the recipient. Mycelial plugs of donor and recipient strains were placed 1 cm apart on PDA and incubated at 28 °C for 7 days. Mycelial agar plugs were re-cultured from the edge of the recipient strain (furthest away from the donor strain) to obtain virus-containing mycelial derivatives. The presence of viral dsRNA in each mycelial derivative was determined.

### 2.7. Measurement of Phenotypic Traits

Fungal morphological characterization: The identified strains of *Fusarium* were cultured at 28 °C in the dark on PDA medium for 7 days. To compare the phenotypes of virus-infected strains, they were cultured under alternating light and dark conditions at 28 °C on PDA medium for 14 days.

Quantification of microconidia: After 14 days of alternating light and dark cultivation at 28 °C on PDA medium, the microconidia on the colony were washed off with sterile water and counted using a hemocytometer under a microscope.

Virulence assay: The inoculation method for shoot rot disease was modified slightly based on the method described by Wang et al. [31]. The syringe needle inoculation method was used. The *Fusarium* strains to be inoculated were cultured on PDA plates for 7 days. The spores were washed off with sterile water, and the mycelia were filtered through four layers of sterile gauze. The spore concentration of each strain was adjusted to 1 × 10^4^ spores/mL under a microscope. During inoculation in sugarcane fields, 500 μL of spore suspension was injected into the sugarcane at 2 cm above the growth point of the 5–6 leaf stage using a 1 mL micro-syringe. Each *Fusarium* strain was inoculated onto 20 sugarcane plants (with 3 replicates). The inoculated sugarcane plants were observed weekly, and the disease development was recorded. At the end of the incubation period, the virulence of the *Fusarium* strains was evaluated using the disease severity index (DSI) of the sugarcane, as described by Wang et al. [31].

## 3. Results

### 3.1. Identification of Fusarium Species Containing FsHV1

During the period from 2012 to 2020, we isolated 214 strains of *Fusarium* from the leaf tissue and rhizosphere of sugarcanes showing Pokkah boeng symptoms for virus detection. The RNA-seq of the pooled RNA of these 214 strains revealed that a total of six mycoviruses, Fusarium sacchari hypovirus 1 (FsHV1), Fusarium sacchari alphavirus-like virus 1 (FsALV1), Fusarium sacchari chrysovirus 1 (FsCV1), Fusarium andiyazi mitovirus 1 (FaMV1), Fusarium andiyazi mitovirus 2 (FaMV2), and Fusarium sacchari mitovirus 1 (FsMV1), were present. We performed RT-PCR on these 214 *Fusarium* strains, including 122 strains of *F. sacchari*, 47 *F. andiyazi*, 24 *F. proliferatum*, 6 *F. oxysporum*, 5 *F. verticillioides*, 5 *F. globosum,* 2 *F. fujikuroi*, 2 *F. solani*, and 1 *F. delphinoides*, with primer pairs specific to these viruses to identify their host range. A total of 12 *Fusarium* strains were found to carry only FsHV1 (Figure 1A,B) and they were selected for further study. The colony morphology of these strains varied from white fluffy to pink fluffy aerial mycelia but with an orange pigmentation secreted into the medium (Figure 1C). Among them, nine *Fusarium* strains (FS1, FS65, FS66, FS9, FS18, FSA1, LZ8, LZ14, and FZ06) exhibited abundant mycelia on PDA, initially white and later producing a light-yellow pigment (Appendix A). The macroconidia were relatively slender, slightly fan-shaped, and thinly wedge-shaped, with curved tips and usually three septa (Appendix A). The microconidia were oval, mainly 0-septate, with a few being 1- or 2-septate (Appendix A). The aerial mycelium presented as pseudo-heads (Appendix A), consistent with the description of *F. sacchari* [2,32]. Two *Fusarium* strains (FS3 and LZ12) exhibited abundant mycelia on PDA, initially white and later pigmented (Appendix A). The macroconidia were generally transparent, thin-walled, and slightly curved, with three to six longitudinal septa (Appendix A). The microconidia were oval, with flat bases and usually 0-septate (Appendix A). The aerial mycelium presented as pseudo-heads (Appendix A) or longer chains (Appendix A), consistent with the description of *F. andiyazi* [4,33]. The morphological characteristics of one *Fusarium* strain (LZ1) were as follows: white colonies on PDA (Appendix A) and macroconidia produced on CLA were wide, slightly curved, with round ends, three to seven septa (Appendix A), and abundant in creamy spore cysts (Appendix A). The microconidia were oval, elliptical, or kidney-shaped, 0- or 1-septate microspores (Appendix A) formed in circular pseudo-heads of the aerial mycelium (Appendix A), consistent with the description of *F. solani* [34]. Phylogenetic trees were constructed using three genes: translation elongation factor (TEF1), DNA-directed RNA polymerase II largest subunit (RPB1), and second-largest subunit (RPB2) genes (Appendix A). The results further confirmed the identification of the 12 *Fusarium* strains, with 9 strains identified as *F. sacchari*, 2 strains as *F. andiyazi*, and 1 strain as *F. solani*. This suggests that FsHV1 can spread among at least three *Fusarium* species in fields, distributed across multiple geographical locations including Fuzhou, Fujian, Fusui, Guangxi, and Longzhou, Guangxi (Appendix A).

### 3.2. Comparative Analysis of Full-Length Sequences of 12 FsHV1 Variants

To obtain the full-length sequence of the FsHV1 virus, 10 fragments were amplified and cloned, along with 5′RACE and 3′RACE (Appendix A). The full-length sequences of 12 FsHV1 variants were obtained by means of sequencing and assembly. The analysis of the FsHV1 genome sequences revealed that all 12 FsHV1 variants had a single large ORF, with 12,774 nucleotides encoding 4258 amino acids. The names and genome sizes of the 12 FsHV1 variants are listed in Table 1.

Phylogenetic relationships among members of the hypoviruses based on complete genome sequences were analyzed according to the neighbor-joining method. The results showed that 12 FsHV1 variants were clustered with Wuhan insect virus 14, Fusarium graminearum hypovirus 1, and Cryphonectria hypovirus 1, which belong to the genus *Alphahypovirus* in the family *Hypoviridae* (Figure 2A). The nucleotide sequence homology among the 12 FsHV1 variants ranged from 98.6% to 99.9% and 98.70% to 99.9% amino acid (see Appendix A). The Fujian isolate FsHV1-FZ06 and the Guangxi isolate FsHV1-FS1 with the highest similarity differed by only 19 nucleotides and six amino acids. In contrast, isolates FsHV1-LZ12 and FsHV1-FS66, had the lowest similarity, with 170 nucleotide differences resulting in 60 amino acid changes. The phylogenetic tree constructed based on the full-length nucleotide sequences of the 12 isolates also showed similar results (Figure 2B).

There were five regions (I~V) with a relatively high number of variable amino acids among the 12 FsHV1 variants (see Appendix A). Different virus variants may encode different amino acids at a single site, such as alanine (A) encoded by six variants and threonine (T) encoded by six variants at the 225th amino acid position. A total of four amino acids were encoded at the 708th position, including histidine (H), tyrosine (Y), cysteine (C), or arginine (R). The number of A bases in the PolyA tail of the 12 FsHV1 variants ranged from 15 to 31.

### 3.3. Comparative Analysis of the Sequences of Six FsHV1-Defective RNAs

We identified the loss of a genomic fragment in 6 of the 12 virus variants using RT-PCR, indicating the formation of defective RNAs (Figure 3A). Sequence alignment indicated that all of these deletions occurred in the genome region 3788 to 5739, varied slightly from the start or end positions, and a total of four types of deletion were found (Figure 3B). Of interest, all these deletions were in-frame deletions, i.e., the deletions did not introduce new proteins. We compared the defective sequences of FsHV1 in multiple spores of the same FS1 strain and found only one type of defective viral dsRNA (see Appendix A).

The RT-PCR results showed that when only WT viruses were present in the *Fusarium* strains, such as FsHV1-FS3 or FsHV1-FS18, the total virus titers were 5195 and 3888 copies, respectively. However, in the presence of defective RNAs such as FsHV1-FS65 or FsHV1-FS9, the total virus titers increased by approximately 10-fold to 16,187 and 14,818 copies, respectively, while the titers of the wild type viral dsRNA (WT viral dsRNA) decreased to 1338 and 1370 copies, respectively (Figure 3C). Defective viral dsRNAs were only detected in *F. sacchari* strains and not in *F. andiyazi* or *F. solani* strains. To determine whether the generation of defective RNAs was influenced by the *Fusarium* host or the virus itself, we generated a FsHV1-free *F. sacchari* strain FZ06-VF by screening single-spore-derived colonies. Then, the 12 FsHV1 variants were transferred into FZ06-VF via anastomosis, ensuring that all virus variants had the same host background. Conversion rates varied among the virus variants (see Appendix A). We tested the virus-converted FZ06-VF using RT-PCR and found that only WT FsHV1 dsRNA but not defective viral dsRNA was detected in the recipient strains (Appendix A).

### 3.4. FsHV1 Virus Reduces Sporulation, Pigmentation, and Virulence of the Host Fungus

To investigate the effect of FsHV1 on *Fusarium* species, we compared the biological characteristics of 12 virus variants on the same background, i.e., introduced the virus from the original strain into the same host strain FZ06-VF. After 14 days of alternating light and dark conditions on the PDA medium, the FZ06-VF strain plates exhibited a deep purple pigmentation, while most of the infected strains, except for FsHV1-LZ12, produced little pigmentation (Figure 4A). FsHV1 did not affect the mycelial growth rate on the PDA. The dsRNAs extracted from 12 FsHV1 acceptor strains were electrophoresed in 1.2% (w/v) agarose gels, and bands between 10 and 20 kb were observed (Figure 4B). The RT-PCR detection of FsHV1 was conducted with specific primer pairs FsHV1-DefectF/R (Figure 4C).

When comparing spore production on PDA after 14 days, the non-virulent strain produced 12.75 × 10^8^ spores, while the virulent strains produced between 5.4 × 10^8^ and 6.9 × 10^8^ spores, representing an approximately 50% reduction in spore production (Figure 5A).

To determine if the FsHV1 virus affects the infection ability of the host *Fusarium* species, we conducted a virulence comparison between the FsHV1 non-virulent strain FZ06-VF and 12 virulent *Fusarium* strains on sugarcane plants using the syringe inoculation method. Sterile distilled water was used as a negative control. Each strain was inoculated onto 20 plants (with three replicates), and observations and statistics were recorded every 7 days. The evaluation criteria for disease severity were based on the disease severity index (DSI). The results showed that the DSI reached its peak in week 3 for all inoculations on sugarcane. At this time, the DSI of the non-virulent FsHV1-VF strain was the highest, at 29.33. The DSI for strains carrying FsHV1-FZ06, FsHV1-FS1, FsHV1-FS9, FsHV1-FS65, FsHV1-FS66, and FsHV1-LZ8 ranged from 12.33 to 17.66. The DSI for strains carrying FsHV1-FSA1, FsHV1-FS3, FsHV1-FS18, FsHV1-LZ1, FsHV1-LZ12, and FsHV1-LZ14 ranged from 3.33 to 8.66 (Figure 5B). Sugarcane plants infected with FsHV1-VF spores exhibited more severe symptoms when compared to those infected with FsHV1-containing spores (Figure 5C). Different viruses significantly reduced the virulence of the host, and the strain carrying FsHV1-FSA1 exhibited the lowest virulence.

## 4. Discussion

In this study, a total of 214 *Fusarium* strains representing seven species were tested for the presence of FsHV1. The results showed that 12 strains of *Fusarium* (9 strains of *F. sacchari*, 2 strains of *F. andiyazi*, and 1 strain of *F. solani*) were positive only for FsHV1, with a detection rate of 5.60%. The strains were collected from various locations in Fujian and Guangxi, indicating a relatively wide spread of FsHV1. Mycoviruses can readily be transmitted vertically via fungal spores, but horizontal transmission via hyphal fusion may be limited by vegetative incompatibility among different incompatibility of the same species or different species [35]. Thus, vegetative incompatibility is a major factor hindering the spread of fungal viruses [36]. Our findings that FsHV1 were able to transmit FsHV1 from *F. sacchari*, *F. andiyazi*, or *F. solani* to a virus-free *F. sacchari* strain through hyphal fusion under laboratory conditions echo the fact that FsHV1 was present in *F. solani* isolated from soil and *F. sacchari* and *F. andiyazi* isolated from sugarcane leaves, suggesting a possibility that that FsHV1 could also be horizontally transmitted among different species of *Fusarium* through hyphal fusion in the field. Previously interspecific transmission of Sclerotinia sclerotiorum virus 1 (Ssv1) from *Sclerotinia sclerotiorum* to *Sclerotinia minor* and FaAV1 from *F. graminearum* to ten other *Fusarium* species through hyphal fusion was reported [37].

It was reported that when defective viruses were passaged or transmitted to new hosts, their genome size and structure changed [38], resulting in different populations of defective viral dsRNA in different *Fusarium* species. Six of the twelve FsHV1 variants contained defective viral dsRNA in this study, but only a single type of defective viral dsRNA was present in a virus/host combination, suggesting that there may be a specific mechanism for splicing the viral dsRNA. Moreover, different types of deletions were all in-frame deletions, so that the deletion would not result in new protein species. Of interest is that the deletions do not occur in the essential regions, e.g., the conserved functional domains of PeP C7, DUF3525, RdRp, and DEAH-box or Hrp8. Therefore, whether or not the defective viral dsRNA are dead or alive remains unknown. The presence of defective viral dsRNA was accompanied by the sharp accumulation of total viral dsRNA, but it reduced the WT viral dsRNA accumulation, suggesting an interference effect of the defective viral dsRNA. Similar results have been reported for Rosellinia necatrix partitivirus 2 (RnPV2), where a defective virus affects the replication of its parent virus [39]. In animal cells, defective virus genomes may serve the function of interfering with the replication of the WT virus by monopolizing virus polymerases or competing for structural proteins [40,41]. The existence of defective viral dsRNA has also been reported in the family *Hypoviridae*, but their biological function remains unknown [42,43,44].

In *Fusarium* spp., it has been reported that Fusarium graminearum virus 1 (FgV1) from the *Hypoviridae* family infects *F. graminearum*, leading to reduced mycelial growth, increased pigment production, decreased trichothecene mycotoxin production, and weakened host virulence [22]. Similarly, Fusarium equiseti partitivirus 1 (FePV1) can decrease the growth rate, biomass, and virulence of its host [45]. Fusarium oxysporum f. sp. dianthi virus 1 (FodV1) was associated with the reduced growth of the colonies, diminished conidiation, and significantly reduced virulence of its fungal host. Likewise, *Fusarium* strains carrying FsHV1 variants exhibited reduced sporulation and virulence. Among the virus variants, FsHV1-FSA1-infected *Fusarium* yielded the lowest virulence toward sugarcane plants but still produced a reasonable number of spores (Figure 5), possessing a potential for biocontrol against PDB. On the viral side, little variation in the nucleotide sequences (Appendix A), yet significant differences in virulence attenuation among the FsHV1 variants offer the opportunity to dissect the key motifs or amino acids responsible for the alteration of the phenotypic traits of the host fungus once an infectious clone is developed. Equally important is the observation that FsHV1 could be transmitted between species of *Fusarium*, a characteristic essential for a broader spectrum of pathogen targets.

## 5. Conclusions

The variants of FsHV1 obtained from three *Fusarium* species in this study were highly similar in genome sequence. These variants were able to overcome the incompatibility barriers between *Fusarium* species and horizontally transmit within strains of the same species or between species of *Fusarium*. FsHV1 infection affected the phenotypic traits of its host fungus, including the suppression of sporulation and attenuation of virulence. Defective viral dsRNAs were found in six fungal strains and the presence of defective dsRNA increased the total viral dsRNA but reduced the WT virus in the cells. The discoveries reported lay a new foundation for future progress toward elucidating the mechanism of virus–host interaction and developing viral biocontrol agents against sugarcane PBD.

## Figures and Tables

**Figure 1 viruses-16-00608-f001:**
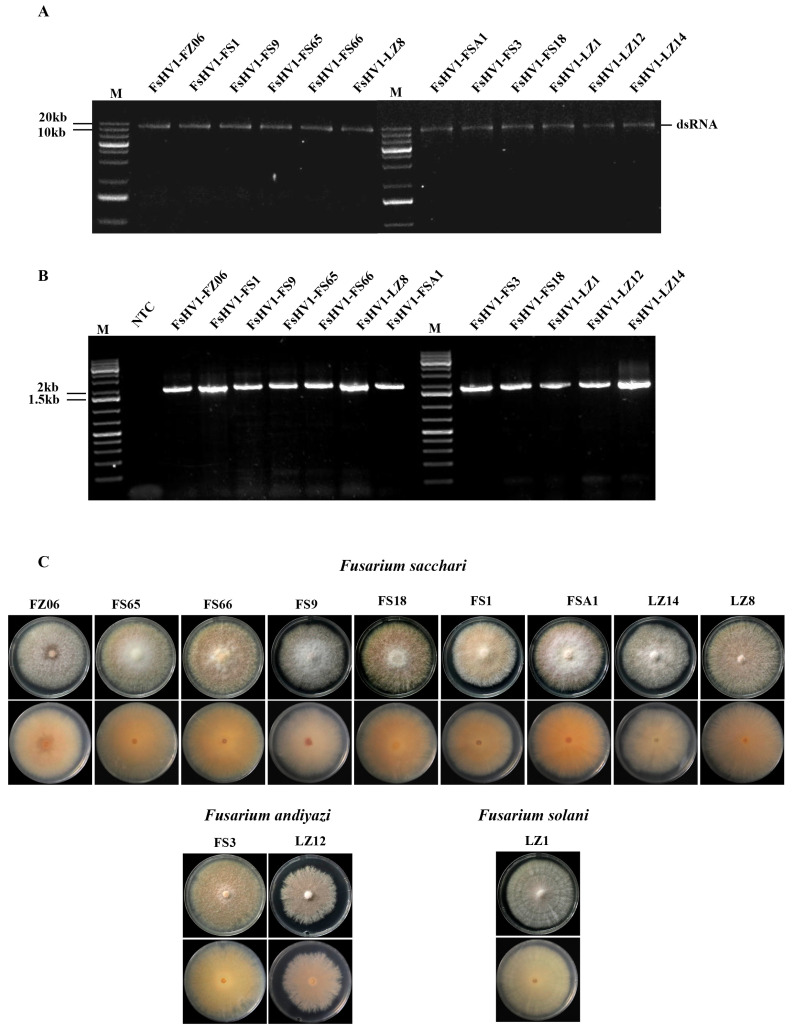
Isolation of *Fusarium* strains carrying FsHV1. (**A**) dsRNA molecules purified from 12 *Fusarium* strains and separated with 1.2% agarose gel electrophoresis. All samples were treated with DNase I and S1 Nuclease. Lane M, GeneRuler 1 kb Plus DNA Ladder. (**B**) RT-PCR-amplified RdRp of FsHV1 derived from 12 strains of *Fusarium* isolate were subjected to 1% agarose gel electrophoresis. The target size is 1700 bp. Lane M, GeneRuler 1 kb Plus DNA Ladder (Thermo Scientific); lane NTC, No template control. (**C**) Colony morphology of 12 strains of *Fusarium* with FsHV1 after 7 days of culture on PDA in the dark.

**Figure 2 viruses-16-00608-f002:**
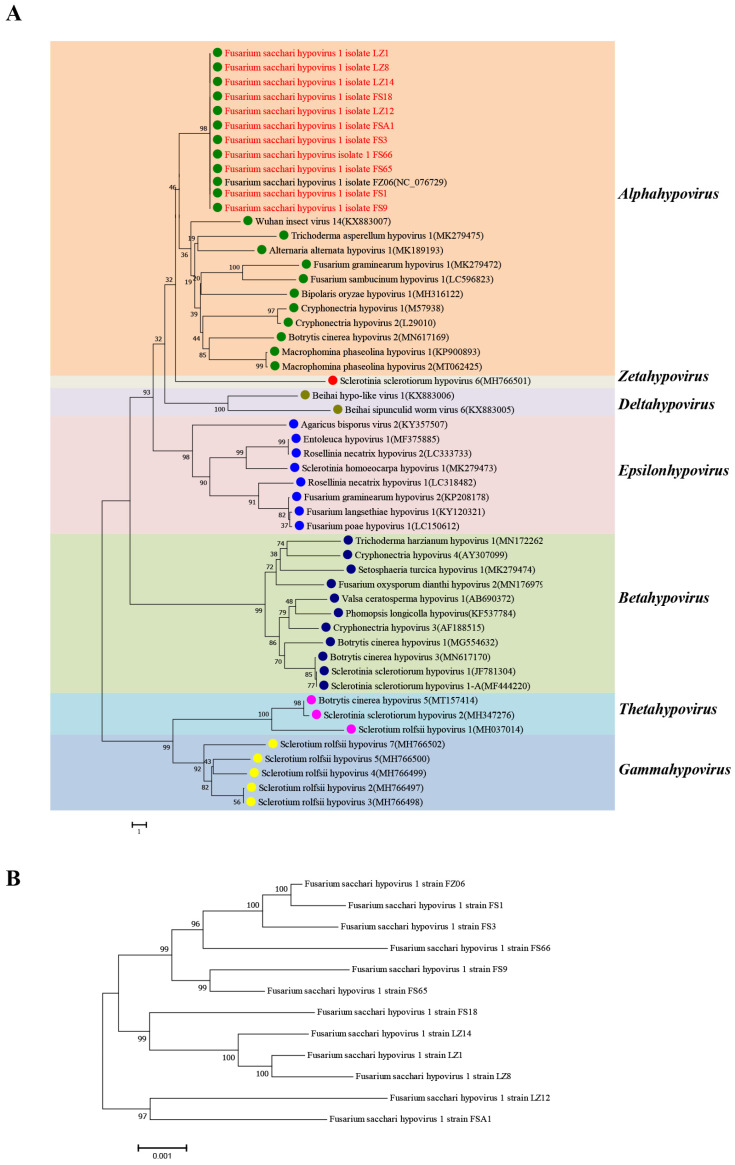
Phylogenetic analysis of FsHV1. (**A**) Phylogenetic relationships of hypoviruses. A neighbor-joining phylogenetic tree based on complete genome sequences and the statistical analysis of branches was evaluated by means of bootstrap support (1000 replicates). Species belonging to the seven genera are identified by colored dots at branch tips, and the 12 FsHV1 variants are shown in red font. Names and GenBank accession numbers of the other 41 hypoviruses are shown in the tree. (**B**) Phylogenetic relationships among the FsHV1 variants. Neighbor-joining phylogenetic tree based on the full-length nucleotide sequences of 12 variants among the FsHV1 variants. The scale bar at the lower left represents a genetic distance. Bootstrap values obtained with 1000 replicates are represented on branches.

**Figure 3 viruses-16-00608-f003:**
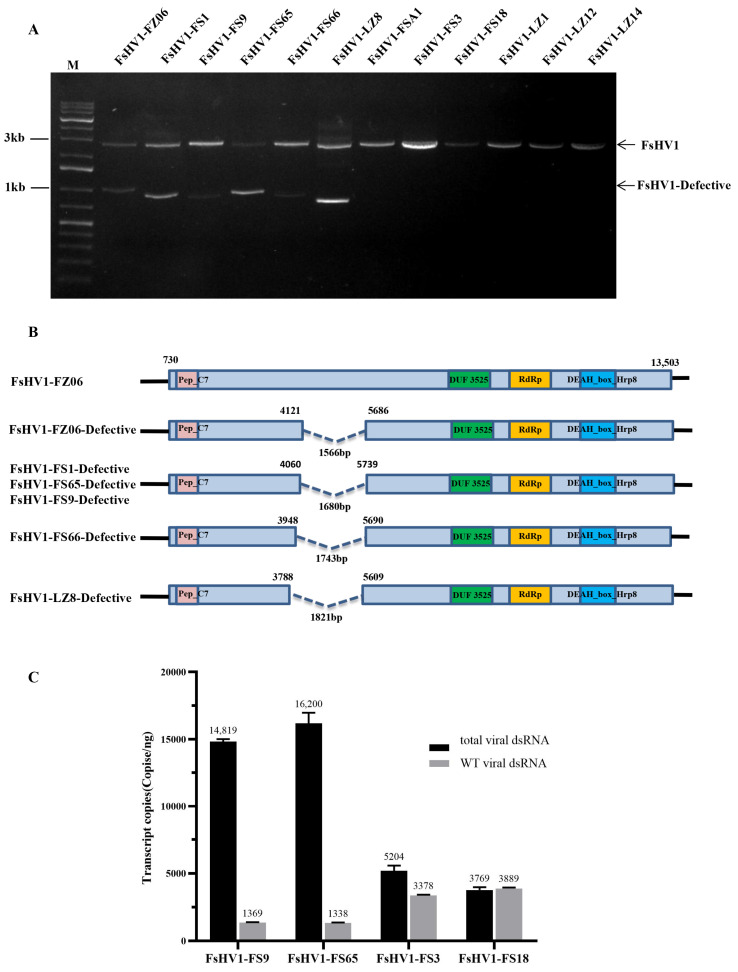
Identification of the FsHV1-defective RNA. (**A**) Confirmation of the presence of the FsHV1-Defective RNA by means of specific RT-PCR using primer pairs FsHV1-DefectF/R. The target size of the WT viral dsRNA is about 2500 bp and the target size of the defective viral dsRNA is about 800 bp. Lane M, GeneRuler 1 kb Plus DNA Ladder. (**B**) Schematic representation of the generation of six FsHV1 defective dsRNAs. FsHV1-FZ06 is a WT virus and has a large putative ORF that is represented by the shallow blue box. --- represents the region of deletion. (**C**) Quantification of FsHV1 dsRNA. The quantity of the WT viral dsRNA was determined by means of qRT-PCR of the deletion region, while the quantity of the total viral dsRNA was determined by means of qRT-PCR of the RdRp region.

**Figure 4 viruses-16-00608-f004:**
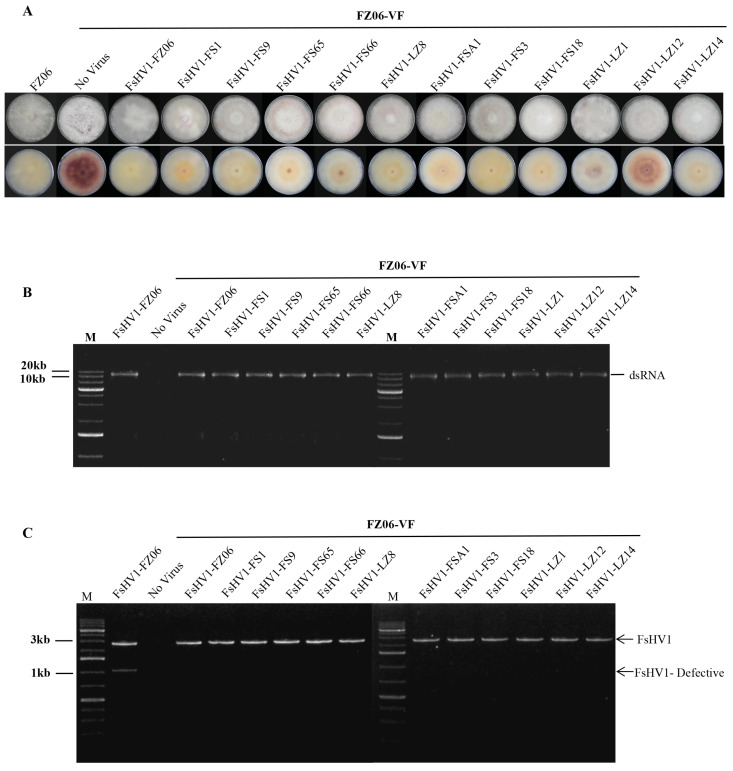
Detection of the horizontal transmission of each of the FsHV1 variants to the FZ06-VF strain. (**A**) Colony phenotype of strains cultured on PDA for 14 days in alternating light and dark. (**B**) dsRNA extracted from the FZ06-VF strain. The FZ06 strain as a positive control. (**C**) RT-PCR detection of FsHV1 infection from the FZ06-VF strain using specific primer pairs FsHV1-DefectF/R. FsHV1-FZ06 as a positive control. Lane M, GeneRuler 1 kb Plus DNA Ladder; lane NTC, No template control.

**Figure 5 viruses-16-00608-f005:**
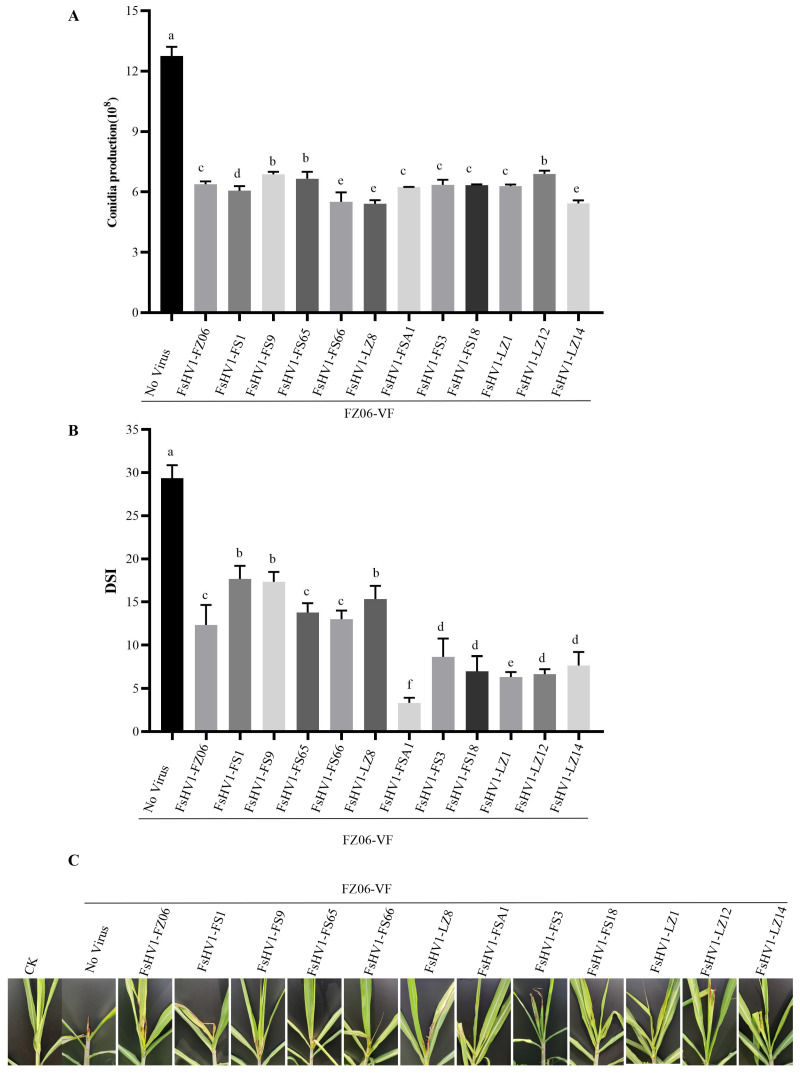
The impact of FsHV1 infection on the biological properties of *F. sacchari*. (**A**) Conidiation. Spore production was expressed as an average per 120 mm Petri dish. Error bars represent the standard deviations of the means from three biological replicates. (**B**) Virulence quantification. Disease severity index (DSI) was calculated from three independent assays, with 20 seedlings. (**C**) Symptoms of sugarcane plants. For virulence tests, conidial suspensions were inoculated onto sugarcane plants. CK, water-sprayed control plants. Error bars represent standard deviations (SD) (n = 3) and different letters (a–f) indicate a significant difference (*P* < 0.05).

**Table 1 viruses-16-00608-t001:** Source and molecular characteristics of the FsHV1 variants used in this study.

No.	Name of FsHV1 Isolate	Abbreviation	Genome Size	5′ UTR	3′ UTR
1	Fusarium sacchari hypovirus 1 isolate FZ06	FsHV1-FZ06	13,975 bp	735	466
2	Fusarium sacchari hypovirus 1 isolate FS65	FsHV1-FS65	13,983 bp	735	474
3	Fusarium sacchari hypovirus 1 isolate FS66	FsHV1-FS66	13,980 bp	735	471
4	Fusarium sacchari hypovirus 1 isolate FS9	FsHV1-FS9	13,979 bp	735	470
5	Fusarium sacchari hypovirus 1 isolate FS18	FsHV1-FS18	13,981 bp	735	472
6	Fusarium sacchari hypovirus 1 isolate FS1	FsHV1-FS1	13,975 bp	735	466
7	Fusarium sacchari hypovirus 1 isolate FSA1	FsHV1-FSA1	13,966 bp	735	458
8	Fusarium sacchari hypovirus 1 isolate LZ14	FsHV1-LZ14	13,980 bp	735	471
9	Fusarium sacchari hypovirus 1 isolate LZ8	FsHV1-LZ8	13,982 bp	735	473
10	Fusarium sacchari hypovirus 1 isolate FS3	FsHV1-FS3	13,982 bp	734	474
11	Fusarium sacchari hypovirus 1 isolate LZ12	FsHV1-LZ12	13,967 bp	735	458
12	Fusarium sacchari hypovirus 1 isolate LZ1	FsHV1-LZ1	13,971 bp	735	462

## Data Availability

The sequences reported in the present manuscript have been deposited in the GenBank database under accession numbers PP397082-PP397092 (*RPB1*), PP429820-PP429841 (*TEF-1α* and *RPB2*), and PP464954-PP464964 (FsHV1 variants).

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
