# Peer review of "Fusarium sacchari hypovirus 1, a Member of Hypoviridae with Virulence Attenuation Capacity in Phytopathogenic Fusarium Species"

_viruses, 2024, doi:10.3390/v16040608_

Round 1
Reviewer 1 Report
Comments and Suggestions for Authors
The authors claim to report a ‘novel’ hypovirus in the present study, however, the BLAST results indicate that the virus is a variant of the previously reported Fusarium sacchari hypovirus 1 (NCBI Reference Sequence: YP_010800181.1) from strain FZ06 (Yao et al., 2020, Frontiers in Microbiology). The virus FsHV1 (this study) shares 99.1% protein identity with Fusarium sacchari hypovirus 1 (YP_010800181.1) thus it is a variant of the same virus published by the same group. The authors are not giving any information in the paper about this high percentage of identity to the previously reported virus and thus the use of the word ‘novel’ is misleading.
Also in the phylogenetic analysis, the previously reported virus has been included as a novel virus and not as a reported virus with its existing accession number from GenBank.
The title of the paper should be modified and also in the text it should be clearly stated that the viruses are not novel but a variant and the study is investigating the biological properties of the previously reported virus. All sections of the paper should be modified accordingly.
In the result section, the information about the comparative analysis of the sequences and their defective RNAs could be shortened.
More focus should be given to the biological properties of the viruses in both the results and discussion sections.
Other comments in the text are in the attached file.

Comments on the Quality of English LanguageEnglish language quality could be improved.
Reviewer 2 Report
Comments and Suggestions for Authors
This manuscript describes the identification and characterization of a hypovirus from Fusarium species infecting sugarcane in China.
-The manuscript needs improvement on the writing. My comments/corrections are added to the pdf file (attached).
-No description about Pokkah boeng disease in the introduction section
- It is not described whether the defective RNA virus has in-frame ORF or deletion introduce new protein sequence or early translation termination.
- Figure 3C, it is better to use northern blot detection of asses the effect of defective RNA virus on WT virus accumulation
- Since the sequence similarity among 12 virus isolates are very high (>99%), analysis of genetic recombination is unnecessary.
-It is not clear whether 12 fungal strains only contain FsHV1 as dsRNA isolation can not verify this. Thus, it can not be concluded that in horizontal transmission experiment to non-virulent FZ06-VF strains, the reduced pathogenicity is due to FsHV1.
- Figure 6, the pictures of sugarcane plants infected with the fungi need to be presented.

Comments on the Quality of English LanguageWriting needs to be improved
Round 2
Reviewer 2 Report
Comments and Suggestions for Authors
All coments have been adressed.